# Cellulolytic Aerobic Bacteria Isolated from Agricultural and Forest Soils: An Overview

**DOI:** 10.3390/biology13020102

**Published:** 2024-02-05

**Authors:** Angélica Bautista-Cruz, Teodulfo Aquino-Bolaños, Jessie Hernández-Canseco, Evangelina Esmeralda Quiñones-Aguilar

**Affiliations:** 1Instituto Politécnico Nacional, CIIDIR-Oaxaca, Hornos 1003, Santa Cruz Xoxocotlán 71230, Oaxaca, Mexico; taquino@ipn.mx; 2Doctoral Programme in Conservation and Use of Natural Resources, Instituto Politécnico Nacional, CIIDIR-Oaxaca, Hornos 1003, Santa Cruz Xoxocotlán 71230, Oaxaca, Mexico; jhernandezc2014@alumno.ipn.mx; 3Laboratorio de Fitopatología de Biotecnología Vegetal, Centro de Investigación y Asistencia en Tecnología y Diseño del Estado de Jalisco, A.C. Camino Arenero 1227, El Bajío del Arenal, Zapopan 45019, Jalisco, Mexico; equinones@ciatej.mx

**Keywords:** *Bacillus*, hydrolysis capacity, crystalline cellulose, functional characterization, genomic sequencing

## Abstract

**Simple Summary:**

Lignocellulose, consisting of cellulose, hemicellulose, and lignin, constitutes 60% of Earth’s biomass and plays a critical role in the carbon cycle. Abundantly found in plant leaves and stems, cellulose undergoes biodegradation predominantly by cellulolytic microorganisms that produce cellulases. This process is particularly vital for the breakdown of crystalline cellulose in plant cell walls. The effective degradation of cellulose in natural environments hinges on the accurate identification of truly cellulolytic bacteria. This review compiles and analyzes data from the past 11 years on such bacteria, derived from forest and agricultural soils, and offers insights into the functions of cellulolytic bacteria and their cellulase enzymes.

**Abstract:**

This review provides insights into cellulolytic bacteria present in global forest and agricultural soils over a period of 11 years. It delves into the study of soil-dwelling cellulolytic bacteria and the enzymes they produce, cellulases, which are crucial in both soil formation and the carbon cycle. Forests and agricultural activities are significant contributors to the production of lignocellulosic biomass. Forest ecosystems, which are key carbon sinks, contain 20–30% cellulose in their leaf litter. Concurrently, the agricultural sector generates approximately 998 million tons of lignocellulosic waste annually. Predominant genera include *Bacillus*, *Pseudomonas*, *Stenotrophomonas*, and *Streptomyces* in forests and *Bacillus*, *Streptomyces*, *Pseudomonas*, and *Arthrobacter* in agricultural soils. Selection of cellulolytic bacteria is based on their hydrolysis ability, using artificial cellulose media and dyes like Congo red or iodine for detection. Some studies also measure cellulolytic activity in vitro. Notably, bacterial cellulose hydrolysis capability may not align with their cellulolytic enzyme production. Enzymes such as GH1, GH3, GH5, GH6, GH8, GH9, GH10, GH12, GH26, GH44, GH45, GH48, GH51, GH74, GH124, and GH148 are crucial, particularly GH48 for crystalline cellulose degradation. Conversely, bacteria with GH5 and GH9 often fail to degrade crystalline cellulose. Accurate identification of cellulolytic bacteria necessitates comprehensive genomic analysis, supplemented by additional proteomic and transcriptomic techniques. Cellulases, known for degrading cellulose, are also significant in healthcare, food, textiles, bio-washing, bleaching, paper production, ink removal, and biotechnology, emphasizing the importance of discovering novel cellulolytic strains in soil.

## 1. Introduction

In terrestrial ecosystems, lignocellulose, the primary product of photosynthesis, is the world’s most abundant renewable plant resource [1], comprising 60% of Earth’s biomass [2]. It consists of cellulose, hemicellulose, and lignin [3]. 

Forests, spanning over 40 million km^2^ and 30% of Earth’s land [4], are crucial ecosystems and significant carbon sinks with 20–30% cellulose in their leaf litter [5,6]. Agricultural activities, meanwhile, produce roughly 998 million tons of lignocellulosic waste annually [7,8,9]. Therefore, both forests and agricultural practices are key sources of lignocellulosic biomass.

Depending on their origin, plant leaves and stems are composed of 35% to 50% cellulose, 20% to 35% hemicellulose, 10% to 25% lignin, and small amounts of other components [10]. Since hemicellulose forms a recalcitrant complex with lignin, cellulose represents the most accessible biopolymer; therefore, its biodegradation is a key step in the global carbon cycle [11].

Cellulose is a component of the cell wall of green plants, different algae, and oomycetes [12]. It is formed by a linear chain of up to 10,000 glucose molecules; therefore, its biodegradation process begins with its fractionation into smaller units that can penetrate microbial cells and be metabolized [13]. The enzymes responsible for this process are cellulases produced by bacteria, fungi, actinomycetes, and protozoa [14,15]. Cellulases are an enzyme system comprising three types of enzymes that function in a coordinated and synergistic manner [16]: (1) endoglucanases or 1,4-β-D-glucan-4-glucanohydrolases (EC 3.2.1.4), which hydrolyze cellulose chains; (2) exoglucanases, including 1,4-β-D-glucan-glucanohydrolases (cellodextrinases, EC 3.2.1.74) and 1,4-β-D-glucano-cellobiohydrolases (cellobiohydrolases, EC 3.2.1.91), which release cellobiose from the reducing and non-reducing ends of polysaccharide chains, releasing glucose (glucanohydrolases) or cellobiose (cellobiohydrolases); and (3) β-glucosidases (EC 3.2.1.21) that hydrolyze soluble cellodextrins and cellobiose into glucose [17]. Bacteria and fungi are the main biodegraders of cellulose in nature [18]. Some bacteria of the genera *Clostridium*, *Cellulomonas*, *Cellulosimicrobium*, *Thermomonospora*, *Bacillus*, *Ruminococcus*, *Erwinia*, *Bacteroides*, *Acetovibrio*, *Streptomyces*, *Microbispora*, *Fibrobacter*, and *Paenibacillus* can produce cellulases when incubated under aerobic and anaerobic conditions [19,20,21]. Cellulolytic microorganisms play a central role in soil formation processes and the global carbon cycle [22]. Cellulolytic bacteria are identified by qualitatively assessing their ability to hydrolyze cellulose. This process involves culturing the bacteria on media containing artificial cellulose as the sole carbon source and employing dyes such as Congo red or iodine to identify colonies capable of utilizing cellulose. Additionally, some studies measure cellulase activity quantitatively in vitro.

Reports from the international enzyme market indicate that cellulase is the most demanded enzyme, accounting for 20% of the global market [23]. In addition to the involvement of cellulases in the biodegradation of cellulose contained in organic waste, these enzymes have been used in several areas. In healthcare, cellulases are used as a treatment for *Pseudomonas* biofilms as an alternative to antibiotics; in the food and beverage industry, in the production of fruit juices and to improve flavors and fragrances; in the textile industry, in bio-washing and bleaching processes; in the paper industry, for ink removal; in biotechnology, to produce bioethanol; and in the manufacturing of detergents and cleaning and washing products [12,15,24].

Most cellulases used in laboratories and commercial applications are derived from fungi, primarily *Trichoderma*, *Aspergillus*, and *Penicillium*, known for their high enzymatic activity and capacity for hydrolysis [25]. However, bacterial cellulases have aroused great interest due to the ability and natural diversity of bacteria to thrive in various niches; this facilitates the selection of cellulolytic strains resistant to different types of environmental stress [26]. *Bacillus* is one of the most studied genera of bacteria [27]. Species of this genus have been isolated from different environmental niches, allowing them to withstand different types of physical and chemical stress and, therefore, produce alkaliphilic, thermophilic, psychrophilic, acidophilic, and halophilic cellulolytic enzymes [28]. Furthermore, *Bacillus* spp. is among the most appealing bacteria for industrial biotechnology, particularly for cellulase production, because these bacteria are easy to culture and reproduce, have very few nutritional requirements, and produce large amounts of enzymes [26]. Cellulases from *Pseudomonas* and *Sphingomonas* are commercially used in the textile industry [29].

This review presents a comprehensive overview of scientific data on cultivable cellulolytic bacteria in forest and agricultural soils worldwide, covering the past 11 years. Additionally, it provides insights into the general characteristics of cellulolytic soil bacteria and their cellulase enzymes.

## 2. Soil Cellulolytic Bacteria and Methods for Their Identification

Soil harbors high bacterial biodiversity, so bacteria play a crucial role in the main soil processes that regulate the functioning of all terrestrial ecosystems, including biogeochemical cycles that involve C, N, S, and P. Cellulolytic bacteria utilize cellulose as a substrate, converting it into simpler oligosaccharides. These oligosaccharides are then transformed into glucose with the aid of cellulase enzymes [30]. 

The global importance of cellulose biodegradation and its efficient use as a carbon source is paramount [12]. Unlike fossil carbon sources, energy from plant biomass achieves a neutral CO_2_ balance, as the CO_2_ emissions result from carbon previously absorbed by plants. This process does not alter atmospheric carbon levels, so it does not contribute to the greenhouse effect [31].

The accumulation of cellulose in the soil can lead to soil pollution [32]. Therefore, the biodegradation of cellulose by soil microorganisms is crucial. Additionally, the decomposition of organic residues contributes a significant quantity of nutrients to the soil, thereby enhancing soil fertility. This process also helps prevent the depletion of soil organic carbon, which has a positive impact on soil health [33]. 

Previous studies have demonstrated significant effects of environmental conditions on the abundance and decomposition activity of cellulolytic bacteria [34]. Consequently, distinct strains of these bacteria may be identified in varying environments [35]. Many of these strains belong to the phyla Firmicutes, Actinobacteria, Proteobacteria, and Bacteroidetes [36,37].

In general, 16S rRNA gene sequences are used to characterize microbial communities [38,39], but this method has a limited resolution, especially when only short gene sequences (i.e., amplicons) are used [40]. Due to the considerable functional variability within taxa, taxonomic identification per se does not provide reliable information on the metabolic properties of a strain [41]. To this end, the characterization of functional genes is advantageous because it allows inferences to be made about the function of interest. Recent studies employing genome or transcriptome sequencing, along with proteome or metabolome analysis, have initiated a new era in understanding the role of plant cell wall-degrading enzymes in the digestive physiology of insects [42]. Sardar et al. [43] utilized a metatranscriptomic methodology to analyze the soil microbial communities and their cellulases within the gut of *Telodeinopus aoutii*, a tropical millipede. Their research revealed that bacteria predominated as the main producers of carbohydrate-active enzymes (CAZymes). This technique holds potential for characterizing communities of cellulolytic bacteria either present in or isolated from agricultural or forest soils.

Single-cell genomics and transcriptomics provide a reliable context for analyzing assembled genome fragments and gene expression activities at the level of individual prokaryotic genomes. These methodologies are rapidly gaining prominence as vital supplements to cultivation-based, metagenomic, metatranscriptomic, and microbial community-focused research approaches. They enable direct access to data from individual microorganisms, including those belonging to deeply branching phylogenetic groups that currently lack cultured representatives. This advance offers unprecedented insights into the molecular and functional diversity of microbial life [44].

## 3. Methods for Qualitative and Quantitative Determination of Cellulase Activity

Typically, cellulolytic bacteria are identified through the qualitative assessment of their cellulolytic capacity. This involves culturing them in media that contains only artificial cellulose as the carbon source. Dyes, such as Congo red or iodine, are used to detect bacterial colonies capable of utilizing cellulose [45]. The fundamental principle underlying most of these methods is the hydrolysis of a cellulose substrate [46]. Teather and Wood [47], as well as Wood et al. [48], noted the effectiveness of Congo red in tests for hydrolyzed polysaccharides. This dye specifically binds to non-hydrolyzed polysaccharides, allowing clear differentiation between colonies that can and those that cannot utilize cellulose, as evidenced by distinct clearance zones around the former [46].

While numerous studies have documented the use of Congo red agar and various dyes as indicators for monitoring cellulose hydrolysis, it is imperative to recognize that some researchers assert that the capability of hydrolyzing cellulose does not always directly correspond with the bacterial production of cellulase enzymes [16,27].

The breakdown of cellulose into glucose involves a complex synergy of glycosyl hydrolase enzymes. Various bacteria and fungi are capable of degrading cellulose, and their cellulolytic activities can be assessed through both qualitative and quantitative methods. Qualitative approaches, such as the use of Congo red dye, are commonly utilized in preliminary screening studies. However, these methods do not provide information about the quantity of cellulase enzymes produced [49]. In contrast, spectrophotometric methods offer more precision by quantifying the levels of reducing sugars using specific substrates. For instance, carboxymethylcellulose is used as a substrate to evaluate endoglucanase activity, while Avicel cellulose is used to assess exoglucanase activity. Furthermore, Whatman filter paper is employed as a medium for measuring the overall cellulase activity of microorganisms [49].

The filter paper activity (FPA) assay, initially introduced by Mandels and recognized by the International Union of Pure and Applied Chemistry (IUPAC), is the established method for determining overall cellulase activity. This technique assesses the degree of filter paper decomposition [50]. IUPAC specifies the standard for calculating filter paper cellulase units (FPU) as the production of 2.0 mg of reducing sugars, equivalent to glucose, from 50 mg of filter paper. This is equivalent to a 4% conversion rate within a 60 min period [51]. However, the accuracy and consistency of the FPA method in assessing cellulase activity are often compromised by the common lack of β-glucosidase in natural cellulase complexes [52].

The Bradford protein assay, a dye-binding method, is used to determine the protein content of cellulase enzymes. This assay, based on the colorimetric change of a dye in response to varying protein concentrations, is frequently preferred in research applications over other protein assays, such as the widely used Lowry method. The Bradford assay offers several advantages: it is simpler to use, requiring only a single reagent and about 5 min to complete, in contrast to the three reagents and 30–40 min needed for the Lowry assay. Additionally, the stability of the dye–protein complex’s absorbance in the Bradford assay eliminates the need for precise timing, a critical aspect of the Lowry method. Furthermore, the Bradford assay is less prone to interference from compounds that might compromise the accuracy of the Lowry assay. The underlying principle of the Bradford assay is that the absorbance peak of an acidic solution of Coomassie Brilliant Blue G-250 shifts from 465 nm to 595 nm upon protein binding [53,54].

Another common technique for cellulase detection involves quantifying the reducing sugars released during the hydrolysis of cellulose [55,56]. These sugars can be measured using various approaches, including high-performance liquid chromatography (HPLC) [57] or spectrophotometry employing diverse dyes or reagents [58,59,60]. However, this methodology’s primary limitation is its inability to distinguish between cellobiose and glucose, both products of different stages in cellulose hydrolysis. Moreover, cellulase activity has been evaluated by observing the change in viscosity of a cellulose solution when treated with cellulase [61]. A single cellulase unit is defined as the enzymatic activity causing a relative fluidity change of one in 5 min in a specific CMC substrate at 50 °C (pH 4.5) [62]. Viscosimetry is particularly indicative of endocellulase activity, as exocellulase enzymes typically induce minimal or no viscosity changes [63].

The most widely used method for assessing endoglucanase activity is the Miller method, developed in 1959 [64]. This method employs 3,5-dinitrosalicylic acid (DNS) for the analysis. Although the core procedure has remained consistent, various modifications have been made by researchers. Variations are noted in several parameters, including the type and concentration of the buffer (such as 0.05 M sodium citrate at pH 4.8 or 0.1 M sodium acetate at pH 5.0), the concentration of the carboxymethyl cellulose (CMC) substrate (typically ranging from 0.5–2.0%), the volume of DNS reagent used, the amount of biological material in the reaction mixture (ranging from 0.2–1.0 mL), and the incubation conditions, which encompass temperature (30–50 °C) and duration (10–30 min) [16,65,66].

The assessment of exoglucanase activity typically involves the use of microcrystalline cellulose, with Avicel often selected as the substrate [67,68]. The procedure includes mixing 0.5 mL of the enzyme solution with an equal volume of 1.0% Avicel cellulose, suspended in a 0.05 M Tris-HCl buffer at a pH of 8.5. This mixture is then incubated at a temperature of 70 °C for a period of 10 min [67].

β-glucosidase activity can be measured using 4-methylumbelliferyl-β-D-cellobioside (4-MUC) as a fluorogenic substrate, as documented by Chernoglazov et al. [69] and Koubová et al. [70]. Another method to determine β-glucosidase activity involves a spectrophotometric approach. In this method, p-nitrophenyl-β-D-glucopyranoside (pNPG) is used as the substrate, with p-nitrophenol being released as a product of hydrolysis, as described by Grata et al. [49].

Numerous studies have conducted quantitative evaluations of the in vitro cellulolytic capabilities of microorganisms. These assessments typically involve measuring the activity levels of enzymes such as β-1,4-endoglucanases, β-1,4-exoglucanases, and β-glucosidases, in addition to overall cellulase activity. These evaluations are carried out under various environmental conditions to account for differences in pH, temperature, humidity, and salinity [26,45].

Temperature significantly affects bacterial growth and the production of extracellular enzymes by altering the physical properties of the cell membrane [71]. Thus, it is critical to determine this physiological parameter, as bacterial cells perish and their metabolites become damaged above their optimal temperature, while their metabolism becomes inactive below this optimal range [72]. The ideal temperature for growth and cellulase activity varies depending on the bacterial strain and the site of isolation [26]. Goyari et al. [73] found that cellulolytic microorganisms exhibit optimal expression in a culture medium when incubated under conditions that mimic their natural environment. Notably, bacteria from the *Streptomyces* genus have been observed to produce significantly more cellulase at a pH of 5 compared to a pH of 7 [70]. This is particularly relevant for cellulolytic streptomycetes in forest soils, which are often characterized by lower pH levels.

## 4. Truly Cellulolytic Bacteria

As previously mentioned, cellulose is present in the cell wall as elongated submicroscopic structures known as micelles. These structures are organized into larger entities called microfibrils, which in turn are assembled into a highly organized crystalline formation [74]. Crystalline cellulose exhibits a high resistance to enzymatic degradation due to its significant degree of crystallinity [31]. The fibers of crystalline cellulose are closely interconnected by non-covalent hydrogen bonds, leading to an enzymatic hydrolysis rate that is 3 to 30 times slower than that observed in amorphous cellulose regions [75]. Therefore, while amorphous cellulose is rapidly degraded, increased crystallinity in cellulose enhances its resistance to enzymatic breakdown [74].

Factors affecting the recalcitrance of lignocellulosic biomass are strongly interconnected and difficult to separate [76]. These factors can be categorized into (a) structural factors, primarily including cellulose-specific surface area, crystallinity, degree of polymerization, and pore size and volume; and (b) chemical factors, relating to the composition and content of lignin, hemicelluloses, and acetyl groups. Lignin acts as a physical barrier that restricts polysaccharide accessibility by impeding enzyme access to cellulose. Moreover, it can irreversibly bind cellulases and other enzymes during enzymatic hydrolysis due to its hydrophobic structural features, including hydrogen bonding, methoxy groups, and polyaromatic structures [77]. Particle size has been identified as a critical parameter affecting cellulose hydrolysis potential [78,79]. Reducing particle size through processes like milling, grinding, and extrusion can improve the interaction between cellulose and enzymes, break down the compact structure of lignocellulose, and enhance hydrolysis rates [80,81]. The accessible volume of cellulose in lignocellulosic biomass is considered an important factor influencing enzymatic breakdown [82]. The accessibility of pore volumes to enzymes varies depending on their size or shape. As the typical size of a cellulase is around 5.1 nm, only pores larger than 5.1 nm are presumed to be accessible to the enzyme [83]. The estimated half-life of cellulose in its natural form, that is, crystalline cellulose, is millions of years at a neutral pH and in the absence of enzymes [74].

CAZymes have catalytic and carbohydrate-binding modules (or functional domains) that degrade, modify, or create glycosidic bonds [84]. These enzymes were classified into families according to the structure of their domains and grouped in the CAZy database [84] as glycosyl hydrolases (GH), carbohydrate esterases (CE), glycosyl transferases (GT), polysaccharide lyases (PL), or enzymes with auxiliary activities (AA) [85]. Most enzymes with cellulolytic activity are GH, which have been classified into different families based on their amino acid sequence and the resulting protein structures and, consequently, their catalytic mechanism [86]. Enzymes whose cellulolytic activity has been characterized belong to the families GH1, GH3, GH5, GH6, GH8, GH9, GH10, GH12, GH26, GH44, GH45, GH48, GH51, GH74, GH124, and GH148 www.cazy.org (accessed on 15 November 2023) [87]. 

Since cellulases are inducible enzymes, microorganisms synthesize them during their development in lignocellulosic materials [88]. Bacterial strategies are variable in terms of the architecture of cellulolytic enzymes, since they can be secreted as independent catalytic units, forming multi-modular structures called cellulosomes, or as multifunctional proteins that can be found free or in cellulosomes [89,90]. Cellulosomes are bound to the cell membrane and were first described in the anaerobic thermophilic bacterium *Hungateiclostridium thermocellum*, formerly known as *Clostridium thermocellum* [91]. These enzymes are characteristic of anaerobic bacteria, while aerobic bacteria secrete free enzymes [90].

Enzymes of the families GH5, GH6, GH8, GH9, and GH48 are essential in the degradation of crystalline cellulose [31]. Surprisingly, to the extent currently known, all organisms possessing at least enzymes of the family GH48 are truly cellulolytic; that is, they can significantly degrade crystalline cellulose (Figure 1a) [2,31,92]. This makes GH48 a valuable tool for identifying truly cellulolytic bacteria, which cannot be performed using the 16S rDNA gene sequence alone [2]. López-Mondéjar et al. [86] indicated that truly cellulolytic bacteria contain several coding genes for the families GH1 and GH3 that encode β-glucosidases, along with genes that encode endo- and exocellulases from other GH families. In contrast, most bacteria that produce enzymes GH5 and GH9 can hydrolyze soluble (artificial) cellulose substrates such as carboxymethylcellulose but are unable to degrade crystalline cellulose (Figure 1b) [2]. Unlike truly cellulolytic species, these so-called “cellulolytic” bacteria cannot degrade and fully utilize crystalline cellulose (Figure 1). Their cellulases are functional, for example, in plant cells infected with pathogens, for the synthesis of cellulose (in both bacteria and plants) and for purposes other than the metabolization of crystalline cellulose [31].

Only a limited number of bacterial strains have been found to possess more than three genes for β-1,4-endoglucanases, which are essential for the effective degradation of crystalline cellulose [31]. This indicates that while the presence of cellulolytic genes in microbial genomes suggests a potential capacity to degrade cellulose, they do not necessarily confirm the actual production of these enzymes in the presence of lignocellulose [86]. Fortunately, with advancements in technology, gene expression and proteome analyses have become feasible and can provide the required evidence for enzyme production [93,94].

Shamshitov et al. [33] reported that 35 of the 64 bacterial isolates with presumed cellulolytic capacity found in agricultural soils did not produce cellulose-degrading enzymes, although they had shown hydrolysis zones in carboxymethylcellulose agar. López-Mondéjar et al. [86] reported that no typical cellulolytic proteins were found in the proteomes of *Luteibacter* L214 even though this strain was able to grow in microcrystalline cellulose medium and showed cellulolytic activity when grown on filter paper. These authors pointed out that the cellulolytic capacity of this bacterium is supported by the presence of type IV pili that mediate adhesion to cellulose [95,96]. Type IV pilus proteins, called pilins, play an important role in competition with other cellulolytic species for adhesion to cellulose [86]. Despite the presence of common cellulolytic genes in *Actinobacteria* genomes [97], only a few isolates could degrade cellulose [98].

Enzymatic systems in cellulolytic bacteria have been elucidated through sequencing and bioinformatic analyses of the genomes of microorganisms that are capable of degrading plant biomass. This research has provided valuable insights into the diversity of enzymes involved [31]. Approximately half of the bacteria with genes encoding cellulases, hemicellulases, and pectinases are saprophytes, that is, bacteria proficient in degrading dead plant biomass [31]. However, despite the widespread presence of cellulose, only a small fraction of microorganisms are capable of completely degrading it [74].

## 5. Cellulolytic Aerobic Bacteria in Forest Soils

Forests are one of the most important and extensive ecosystems on Earth, covering more than 40 million km^2^ and representing 30% of the world’s surface [4]. Forest ecosystems are important carbon reservoirs, with large amounts of recalcitrant carbon in their soils, especially in temperate forests, which receive tons of litter per hectare each year [5]. Leaf tissue accounts for more than 70% of the litter that falls to the forest floor; the rest comprises stems, small twigs, and propagation structures [99]. It is estimated that between 20% and 30% of forest leaf litter is cellulose [6]. The cellulose content of forests depends on the type of vegetation and the time of the year [100].

Forest ecosystems also provide multiple habitats for bacteria, including soil, plant tissues, streams, and rocks, among others, although bacteria appear to be particularly abundant on the forest floor and soil [101]. The presence of cellulolytic bacteria in the rhizosphere can be explained by the high organic matter content in this zone from lignocellulosic plant and animal residues [102].

Viteri-Florez et al. [100] documented the presence of bacilli with cellulose-degrading capabilities in forest soils at Páramo El Malmo and Páramo El Horizonte, the Iguaque Natural Reserve, the Iguaque-Arcabuco trail, and the Gomeca River basin in the Department of Boyaca, Colombia (Table 1). The cellulolytic bacterial population in these soils ranged between 6.0 and 6.9 × 10^3^ CFU g^−1^. Furthermore, these authors were the first to report evidence of cellulolytic activity in the genus *Erwinia*.

Avellaneda-Torres et al. [45] isolated bacteria with cellulolytic potential from the soils of extreme environments in the Superpáramo, Páramo, and the High Andean Forest in the Nevados National Natural Park, Colombia. These high-mountain ecosystems are considered extreme environments due to their high levels of solar radiation, low atmospheric pressure, and daily temperature changes, found mainly in Colombia, Ecuador, and Venezuela. The soil of these three ecosystems had acidic to highly acidic pH values: Superpáramo (pH 5.9), Páramo (pH 5.5), and the High Andean Forest (pH 4.2). The carbon and nitrogen contents were high, although the Superpáramo soil showed the lowest content of these elements due to the scarce presence of vegetation, where *Calamagrostis* sp. was the dominant species. In total, 74 cellulolytic bacterial isolates were obtained from these soils. Of these, only 25 were selected as the most efficient based on their cellulolytic capacity. The bacterial genera to which these isolates belong are shown in Table 1. Cellulolytic bacteria isolated from Páramo soil showed the highest endoglucanase and total cellulase activity; however, the authors did not report the individual enzymatic activity of each isolate.

López-Mondéjar et al. [86] isolated bacteria with cellulolytic potential from both the soil and forest floor of a temperate oak forest (*Quercus petraea*) in the Czech Republic. They observed cellulolytic activity in approximately 12% of bacterial colonies cultured on plates and stained with Congo red. In total, 115 isolates were obtained, and sequencing identified 42 operational taxonomic units (OTUs) with a similarity of 99%. These OTUs were classified into 22 bacterial genera belonging to the phyla Actinobacteria, Bacteroidetes, Proteobacteria, and Firmicutes. The identified cellulolytic bacteria within these OTUs were more abundant on the forest floor than in the soil. The most prevalent cellulolytic bacterial genera are listed in Table 1. Several isolates demonstrated enzymatic activities related to the biodegradation of cellulose and hemicellulose, especially when cellulose was the sole carbon source.

Khotimah et al. [103] found cellulolytic bacteria in swamp/peat forests in Indonesia (Table 1). The cellulolytic bacterial population ranged from 2.1 × 10^3^ to 5.9 × 10^4^ CFU g^−1^. *Bacillus cereus* showed the highest cellulase activity, 11.17 U mL^−1^. The soil of these forests had a high organic carbon content (56%), low total N (1.88%), high P_2_O_5_, and very acidic pH values. The average moisture content was considered moderate, with 417%. According to these authors, peatland drainage systems were able to modify the condition from anaerobic to aerobic, leading to increased microbial activity.

Tang et al. [104] isolated cellulolytic bacteria from soils from a rehabilitated forest site in Malaysia. This soil had a pH of 4.71 and a cation exchange capacity of 8.13 cmol_c_ kg^−1^. The exchangeable K, Ca, and Mg contents were 0.10, 6.68, and 45.33 cmol_c_ kg^−1^, respectively. Total N content was 0.13%, total organic carbon was 2.60%, and organic matter was 4.48%. These authors reported 14 cellulolytic isolates. The cellulolytic species found are shown in Table 1. The highest cellulase activity of these isolates was 0.089 and 0.077 U mL^−1^. The *Serratia nematodiphila* strain reported in this study produced indole acetic acid (11.39 µg mL^−1^) and solubilized tricalcium phosphate (14.25 µg mL^−1^). These findings suggest that this bacterial strain can improve plant growth and soil fertility in addition to its ability to biodegrade cellulose.

In the mangrove forests of the Bagerhat district in Bangladesh, characterized by soils with an organic carbon content of 1.19 kg/m², a pH of 5.9, and a silt–loam texture, Biswas et al. [105] isolated 17 bacterial strains capable of biodegrading cellulose. Most of these isolates were identified as species within the genus *Pseudomonas*, as detailed in Table 1. The cellulase secretion ability of these isolates varied, reflecting differences in their cellulose degradation capabilities.

In forests of the Montiers experimental site in France, where beech is the dominant tree species, Bontemps et al. [106] found 79 isolates of the genus *Streptomyces* with cellulolytic capacity. *Streptomyces* is a gram-positive filamentous bacterium of the order Actinomycetal. Carboxymethylcellulose is an easy-to-degrade type of cellulose, but *Streptomyces* can also use more recalcitrant forms of crystalline cellulose [94,112]. In fact, some studies show evidence of the co-metabolism of lignocellulosic polymers by several *Streptomyces* spp. [98].

In pristine forests of Patagonia (Argentina) where soil pH ranged from 4.5 to 7.7 and organic matter content varied from 4.24% to 13.96%, Ghio et al. [109] and Ghio et al. [110] isolated cellulolytic bacteria. Among these, gram-positive rods predominated, with a marked prevalence of bacteria of the order Bacillales. The identified cellulolytic species are shown in Table 1, although the isolates with the highest cellulolytic capacity were *Variovorax paradoxus* and *Paenibacillus alvei*.

Ma et al. [108] obtained 81 bacterial isolates capable of biodegrading cellulose from rotten wood samples (weeds, red birch, poplar, alpine rhododendron, and willow) in the Qinling mountains, China. Based on the diameter of the hydrolysis halo using Congo red, only 55 isolates were selected, most of which belonged to the genus *Bacillus* (Table 1). Eight strains showed high cellulase activity. In the forests of Nallamala Srisailam, Kurnool district, Andhra Pradesh (India), Ashwani et al. [111] reported bacteria with the capacity to metabolize cellulose; these bacterial genera are shown in Table 1. The soil of these forests was enriched with plant residues, exhibiting the following characteristics: clay (50%); silt (40%), sand (10%), organic matter (8.0%); total nitrogen (9.0 g kg^−1^), available phosphorus (400 kg ha^−1^), and available K (1580 kg ha^−1^).

Bhagat and Kokitkar [29] isolated 45 cellulase-producing bacteria (based on their ability to discolor Congo red and iodine) from undisturbed forest soils, undisturbed gardens, and open dumps in the Raigad district of Maharashtra (India). Seven isolates showed the highest cellulolytic activity and were selected for characterization and molecular identification. All these isolates were gram-positive bacilli belonging to different species of *Bacillus* (Table 1). The optimal temperature for cellulase production by these seven isolates was 27 °C to 37 °C in a pH range of 6 to 8.

The most abundant cellulolytic bacterial genera in forest soils were *Bacillus > Pseudomonas > Stenotrophomonas > Streptomyces* (Figure 2a). The above studies show that the diversity and abundance of cultivable cellulolytic bacterial communities in the soil of forest ecosystems appears to depend on abiotic factors such as soil properties, environmental conditions, geographic location, seasonality, and biotic factors such as vegetation type, mulch composition, organic matter content, and degree of organic matter breakdown [113,114,115].

## 6. Cellulolytic Aerobic Bacteria in Agricultural Soils

Agricultural activities produce an average of 998 million tons of lignocellulosic waste each year [7,8]. Lignocellulosic crop waste is composed of post-harvest materials that remain on the land, such as straw, roots, peels, stems, and leaves, being one of the most abundant raw materials on Earth [9]. The cellulose levels in agricultural soils depend on the frequency of residue incorporation, a practice which ranges from intermittent to occasionally being overlooked entirely [100].

Dobrzyński et al. [116] and Dobrzyński et al. [117] determined the impact of various agricultural systems and manure fertilization on the abundance of cellulolytic and potentially spore-forming cellulolytic bacteria. The study site was a nearly 100-year-old fertilization experiment, one of the oldest still active field trials, located in the fields of the Institute of Agriculture, University of Life Sciences, Skierniewice, Poland. Treatments included crop rotation with or without legumes, potato, and rye monocultures, as well as treatments with or without manure fertilization. The abundance of cellulolytic and potentially spore-forming cellulolytic bacteria was evaluated using standard microbiological methods, such as the most probable number (MPN) and 16S rRNA gene sequencing. MPN was estimated using the dilution method, and the degradation of filter paper was evaluated macroscopically. In all treatments, the dominant isolates at the order level were Brevibacillales (13.1–43.4%), Paenibacillales (5.3–36.9%), and Bacillales (4.0–0.9%). The dominant families in all analyzed samples were Brevibacillaceae (13.1–43.4%), Paenibacillaceae (8.2–36.9%), and Clostridiaceae (5.4–11.9%). The families Aneurinibacillaceae and Hungateiclostridiaceae increased in the manure fertilization treatments. The authors concluded that the impact of crop management on potentially spore-forming cellulolytic bacteria was negligible, while manure fertilization was a key driver for the potentially spore-forming cellulolytic community. 

Elkhalil et al. [118] obtained 11 cellulolytic bacterial isolates from potato (*Solanum tuberosum* L.) rhizosphere soil (Table 2). This soil had an alkaline pH (8.5) and a clay–silt texture. All isolates were gram-positive, endospore-forming bacilli that showed motility. These characteristics indicated that all isolates belonged to the family Bacillaceae. The cellulase activity of the isolates ranged from 0.80 U mL^−1^ to 2.48 U mL^−1^.

Two cellulolytic bacterial species were isolated from rice crops at Lovely Professional University Campus, India, as detailed in Table 2. Both isolated species are thermophilic and gram-positive. *Geobacillus stearothermophilus* exhibited higher cellulase-specific activity compared to *Bacillus coagulans* [119].

Abdel-Aziz et al. [120] isolated cellulolytic bacteria from agricultural soils in the Menoufia region, Egypt. Twenty-four isolates showed hydrolysis zones on Luria–Bertani agar after Congo red staining. Of these, three isolates were selected for having the largest diameter of hydrolysis zones. These isolates belonged to the genera *Bacillus* and *Klebsiella* (Table 2).

Shamshitov et al. [33] obtained 159 cellulolytic bacterial isolates from agricultural soils with different tillage and cover crop management types. Soil management and identified cellulolytic bacteria are shown in Table 2. The soil was cultivated with peas (*Pisum sativum*) at the time of soil sampling. The soil had a loamy texture; the pH ranged from 6.3 to 6.5; organic carbon content, from 1.40% to 1.69%; and total nitrogen content, from 0.116% to 0.180%. Partial sequencing of the 16S rRNA gene indicated the presence of cellulolytic bacteria represented by members of the phyla Actinobacteria, Firmicutes, Proteobacteria, and Bacteroidetes. *Streptomyces* and *Bacillus* were the most abundant bacterial genera. Regarding enzyme production, 15 strains had endoglucanase activity ranging from 9.09 to 942.41 MUF nanomoles (4- methylumbeliferone) mL^−1^.

Dias et al. [121] employed the C1AC55.07 bacterial strain, isolated from sugarcane (*Saccharum officinarum* L.) soil in Agroindustrial Japungu S.A., Brazil, to identify optimal cellulase production conditions. This strain was identified as *Bacillus* sp. by 16S rDNA gene sequencing. This bacterial strain was evaluated for cellulase enzyme production using an agar medium containing carboxymethylcellulose and incubated at 55 °C for three days. The cellulase activity was detected as transparent zones around colonies using Congo red staining.

In a subhumid tropical area in the province of Manabi, Ecuador, with a mean temperature of 28 °C and about 800 mm of annual precipitation distributed over a six-month rainy season (December to May), Guzmán Cedeño et al. [122] isolated and selected cellulolytic bacteria. Soil management in the study area includes (1) organic agriculture with the constant incorporation of fresh and stabilized organic matter, with 4.06% organic matter content; (2) conventional agriculture with chemical and mechanized agronomic practices, with 1.37% organic matter content, and (3) sugar cane (*Saccharum officinarum* L.) monocrops which have been cultivated for approximately ten years with an evident accumulation of fibrous residues, high moisture retention, and 1.61% organic matter content. The sugarcane plot was a favorable habitat for cellulolytic bacteria, with 41 bacterial isolates, probably because fibrous wastes promoted the growth of cellulolytic microorganisms. The organic agriculture plot showed 26 isolates, probably due to the incorporation of organic fertilizers and conservation practices as part of the ecological management of the soil. The plot under conventional agriculture produced the lowest number of bacterial isolates, probably influenced by the chemical and mechanized practices carried out in the soil, resulting in a low organic matter content and the population dynamics of cellulolytic microorganisms. According to the standard identification method using phenotypic, morphological, and physiological characteristics as differentiation criteria for bacteria, the isolates were bacteria belonging to the genus *Bacillus* spp. (Table 2). 

Máté et al. [123] obtained a gram-negative bacterial isolate called Kb82T from agricultural soil after the harvest of maize (*Zea mays* L.) in Hungary. Cellulose degradation was demonstrated by Congo red staining, and the isolate grew in a medium with carboxymethylcellulose as the only carbon source. The phenotypic characteristics and 16S rRNA gene sequencing supported the conclusion that isolate Kb82T represents a new species in the genus *Flavobacterium* for which the name *Flavobacterium hungaricum* sp. nov was proposed (Table 2). The type of strain for the species is strain Kb82T. GH genes potentially involved in lignocellulose breakdown have been identified. The authors identified 100 GH genes in 32 GH families.

Susilowati et al. [124] isolated cellulolytic bacteria from the rhizosphere of rice (*Oryza sativa* L.). They used a carboxymethylcellulose agar medium using Congo red staining to detect cellulolytic activity. The cellulolytic bacterial isolates belonged to the genera *Streptomyces*, *Bacillus*, *Rhodococcus*, *Pseudomonas*, *Arthrobacter*, *Stenotrophomonas*, *Staphylococcus*, *Acinetobacter*, *Rhodobacter*, *Pantoea*, *Sinomonas*, *Microbacterium,* and *Citrobacter* (Table 2). Bacteria of the genus *Bacillus* exhibited the greatest capacity to produce cellulases.

Trujillo-Cabrera et al. [35] isolated cellulolytic bacteria from the soil of two-century-old *chinampas*. These *chinampas*, located in Paraje Potrero of the Apatlaco Canal in Xochimilco, Mexico City, are traditional agricultural systems. They consist of dead reed fencing and alternating layers of rocks, aquatic vegetation, natural waste, and lake sediments filling the gaps. The man-made origins of the chinampas impart unique soil characteristics, such as elevated moisture, rich organic matter content, high porosity, and saline–alkaline conditions [125]. In the studied *chinampas*, no fertilizers were applied, and at the time of rhizosphere soil sampling, they were grown with alfalfa (*Medicago sativa* L.), chard (*Beta vulgaris* L.), grass (*Arrhenatherum elatius* (Linn.) Pressl), and common sorrel (*Rumex crispus* L.). Bacterial isolates could degrade cellulose in a medium with a pH of 4.5 to 10.0 or supplemented with 1% to 9% NaCl. The highest proportion of cellulolytic isolates and degradation efficiency was recorded at a pH of 6.0. Furthermore, 84.8% of these isolates degraded xylan, and 71.7% degraded Avicel. The phylogenetic analysis of the 16S rRNA genes produced 42 phylospecies within 29 genera belonging to the phyla Actinobacteria, Firmicutes, and Proteobacteria, dominated by the genera *Arthrobacter*, *Streptomyces, Bacillus, Pseudomonas*, *Pseudoxanthomonas,* and *Stenotrophomonas* (Table 2).

The most abundant cellulolytic bacterial genera in agricultural soils were *Bacillus > Streptomyces > Pseudomonas > Arthrobacter* (Figure 2b). Agronomic management critically influences both soil fertility and biological diversity [126]. The widespread adoption of Green Revolution practices, including the application of chemical fertilizers and pesticides [127], has progressively reduced, and in some cases, completely eliminated, the biological components of soil, including cellulolytic microorganisms. Leveraging these microorganisms to accelerate the in situ decomposition of organic waste presents an eco-friendly, cost-effective, and viable approach to recycling crop waste, thereby enhancing soil fertility [33].

## 7. Conclusions and Future Directions

The most abundant cellulolytic bacterial genera in forest soils were found to be *Bacillus*, followed by *Pseudomonas*, *Stenotrophomonas*, and *Streptomyces*. In agricultural soils, the predominant genera were *Bacillus*, *Streptomyces*, *Pseudomonas*, and *Arthrobacter*. Many studies have used artificial cellulose substrates for the qualitative evaluation of bacterial hydrolysis capacity. Although qualitative assessment is valuable, it may not definitively identify truly cellulolytic bacteria. Therefore, there is a need for comprehensive genomic analyses. Genomic sequencing, supplemented with proteomic and transcriptomic tools, can effectively characterize cellulolytic bacteria communities present in, or isolated from, agricultural or forest soils. This genomic information is crucial for understanding the cellulolytic mechanisms, structures, and extracellular enzyme production of these bacteria. Using truly cellulolytic bacteria for the in situ decomposition of organic waste presents an environmentally friendly, cost-effective, and practical strategy for recycling crop waste, thereby enhancing soil fertility. Furthermore, the potential application of these bacteria in various fields, such as sustainable agriculture to promote plant growth, warrants further exploration. Additionally, analyzing the variation, abundance, and diversity of truly cellulolytic microbial communities in soil can provide insights into soil health. The growing global interest in lignocellulosic biomass as a sustainable alternative to fossil carbon resources for the production of second-generation biofuels and biobased chemicals, without compromising food security or exacerbating global warming, highlights the importance of discovering novel cellulolytic strains in soil microorganisms.

## Figures and Tables

**Figure 1 biology-13-00102-f001:**
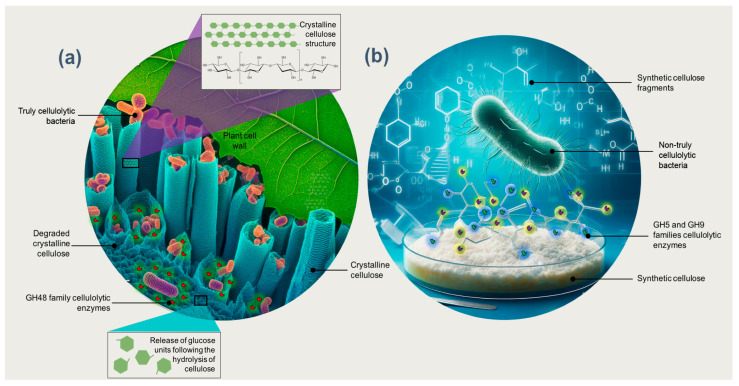
(**a**) Bacteria that are truly cellulolytic, characterized by the production of GH48 family enzymes, possess the capability to degrade crystalline cellulose. (**b**) In contrast, bacteria that are not genuinely cellulolytic and produce enzymes from the GH5 and GH9 families can hydrolyze synthetic soluble cellulose substrates but lack the capability to degrade crystalline cellulose.

**Figure 2 biology-13-00102-f002:**
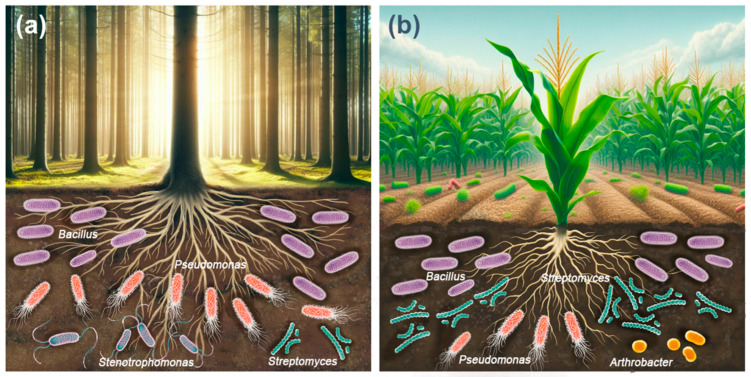
Most abundant bacterial genera with cellulolytic activity in forest soils (**a**) and agricultural soils (**b**).

**Table 1 biology-13-00102-t001:** The most abundant cultivable aerobic bacteria with cellulolytic capacities in forest soils from various regions worldwide.

Isolation Site	Genus/Species	Intensity of Cellulase Activity	Method for Identifying Cellulolytic Bacteria	Carbon Source for Screening Cellulolytic Bacteria	Methods for Analyzing Cellulase Activity	Ref.
Native forests of El Páramo El Malmo El Páramo El Horizonte Iguaque Natural Reserve Iguaque-Arcabuco stretch and Gomeca river basin in the Department of Boyacá, Colombia	*Bacillus* sp. *Erwinia* sp. *Pseudomonas* sp.	+++ ++ ++	Biochemical testing	Carboxymethylcellulose	Qualitative: Congo red assay	[100]
Superpáramo, Páramo, and High Andean Forest in the Nevados National Natural Park, Colombia	*Pseudomonas* *Streptomyces* *Rhodococcus* *Stenotrophomonas* *Variovorax* *Serratia* *Janthinobacterium*	++ +++ + + + + +	16S rRNA gene sequencing	Carboxymethylcellulose	Qualitative: Congo red assay Quantitative: β-glucosidase activity was measured by its absorbance at 405 nm, while endoglucanase and exoglucanase activities were determined using the 3,5-dinitrosalicylic acid (DNS) method. Total cellulase activity was assessed by adding filter paper and phosphate buffer. In all cases the protein content was determined by Bradford method	[45]
Forest floor of a temperate oak forest (*Quercus petraea*), Czech Republic	*Pedobacter* *Mucilaginibacter* *Luteibacter*	++ +++ +	16S rRNA gene sequencing *	Carboxymethylcellulose	Qualitative: Congo red assay Quantitative: Exocellulase and β-glucosidase activities were measured using methylumbelliferol (MUF)-based substrates by mass spectrophotometry proteomics	[86]
Swamp/peat forests, Indonesia	*Bacillus cereus* *Bacillus stratosphericus*	+++ ++	16S rRNA gene sequencing	Carboxymethylcellulose	Qualitative: Congo red assay Quantitative: Endoglucanase activity was analyzed using the DNS (3,5-dinitrosalicylic acid) method, which quantifies glucose levels through spectrophotometry at 540 nm	[103]
Tropical rehabilitated forest soils, Malaysia	*Serratia nematodiphila strain* SP6 *Serratia marcescens* subsp. *sakuensis* *Stenotrophomonas maltophilia* strain KNUC605 *Bacillus thuringiensis* *Stenotrophomonas* sp. Ellin162	+++ ++ ++ ++ ++	Biochemical testing and 16S rRNA gene sequencing	Cellulose microgranular powder	Quantitative: CMCase (carboxymethyl cellulase) activity was determined by quantifying reducing sugars with the Somogyi-Nelson reagent, measured spectrophotometrically at 520 nm	[104]
Mangrove forests of the Bagerhat district, Bangladesh	*Bacillus* sp. *Pseudomonas aeruginosa*	+++ ++	Morphological and biochemical testing, along with 16S rRNA gene sequencing	Carboxymethylcellulose	Qualitative: Congo red assay Quantitative: Endoglucanase and exoglucanase activities were analyzed using the DNS method, which quantifies glucose levels through spectrophotometry at 540 nm	[105]
Montiers Forest experimental site, French	*Streptomyces* sp. strain S2n2 *Streptomyces* sp. strain S8n36	+++ +++	16S rRNA gene sequencing *	Carboxymethylcellulose	Qualitative: Congo red assay	[106]
Mature coniferous forest located in northern Ontario, British Columbia, California and Texas	*Cellvibrio* *Janthinobacterium* *Cytophaga* *Salinibacterium*	+++ ++ + +	16S rRNA gene sequencing	Carboxymethylcellulose	Stable isotope probing (SIP) integrated with amplicon and shotgun metagenomic techniques	[107]
Rotten wood samples, Qinling Mountain in Shaanxi Province, China	*Bacillus subtilis* *B. licheniformis* *B. megaterium* *B. methylotrophicus* *Pseudomonas aeruginosa*	+++ ++ + + +	16S rRNA gene sequencing *	Carboxymethylcellulose or Avicel	Qualitative: Congo red assay Quantitative: FPA (Filter paper activity), activities of CMCase and Avicelase were analyzed using the DNS method, which quantifies glucose levels through spectrophotometry at 540 nm	[108]
Forest soil in the Patagonia region, Argentina	*Variovorax paradoxus* *Paenibacillus alvei* *Pseudomonas jessenii* AMBI2391 *Stenotrophomonas maltophilia CQ1* *Paenibacillus* sp. 61724 *Bacillus* sp. S3.TSA.017 *Bacillus* sp. A2022 *Bacillus arenosi* *Brevundimonas* sp. SOZ3-5041 *Bacillus cereus* SH 01 *Lysinibacillus sphaericus* strain DE4 *Xanthomonas* sp. X1 *Achromobacter xylosoxidans* X96 *Lysinibacillus* sp. KB1	+++ +++ +++ +++ +++ + + + + + +++ + +++ +++	16S rRNA gene sequencing	Carboxymethylcellulose	Qualitative: Congo red assay	[109,110]
Nallamala forest Srisailam, Kurnool District of Andhrapradesh, India	*Bacillus* sp. *Pseudomonas* sp.	Not quantified	Morphological and Biochemical testing	Ashed, acid-washed cellulose powder	Qualitative: Congo red assay	[111]
Dump yards, undisturbed garden soil, and undisturbed forest soil, Raigad district, Maharashtra, India	*Bacillus subtilis* CP053102.1 *Bacillus flexus* NR_113800.1 *Bacillus licheniformis* CP034569.1 *Bacillus paralicheniformis* KY694465.1	+++ + ++ +	16S rRNA gene sequencing	Carboxymethylcellulose	Qualitative: Congo red assay Quantitative: CMCase activity was analyzed using the DNS method, and the quantification of released sugar units was performed following Miller’s protocol.	[29]

* The sequences of the 16S rRNA gene have been archived in the GenBank database. + Level of cellulase activity as reported by the authors: +++ (highest activity), ++ (medium activity), + (low activity).

**Table 2 biology-13-00102-t002:** The most abundant cultivable aerobic bacteria with cellulolytic capacities in agricultural soils across various global regions.

Crop	Site	Genus/Species	Intensity of Cellulase Activity	Method for Identifying Cellulolytic Bacteria	Carbon Source for Screening of Cellulolytic Bacteria	Agricultural Management	Methods for Analyzing Cellulase Activity	Ref.
Potato (*Solanum tuberosum* L.)	Shambat, Khartoum North, Sudan	*Bacillus*	+++	16S rRNA gene sequencing	Carboxymethylcellulose	Not specified	Quantitative: CMCase activity was analyzed using the DNS method, with the quantification of released sugar units carried out following Miller’s protocol	[118]
Rice (*Oryza sativa* L.)	Lovely Professional University Campus, India	*Bacillus coagulans* *Geobacillus stearothermophilus*	++ +++	16S rRNA gene sequencing	Carboxymethylcellulose	Not specified	Quantitative: Total cellulase activity was determined using the DNS method	[119]
Not specified	Menoufia, Egipto	*Bacillus licheniformis,* KT693282 *Bacillus cereus,* KT693283 *Klebsiella oxytoca,* KT693284	+++ ++ +	Morphological and biochemical testing, along with 16S rRNA gene sequencing	Carboxymethylcellulose	Not specified	Qualitative: Congo red assay	[120]
Pea (*Pisum sativum*)	Akademija, Central Lithuania	*Stenotrophomonas rhizophila* *Arthrobacter pascens* *Paenarthrobacter nicotinovorans* *Oerskovia paurometabola* *Terrabacter carboxydivorans* *Agromyces cerinus* *Streptomyces canus* *Streptomyces argenteolus* *Bacillus pumilus* *Bacillus altitudinis* *Bacillus mobilis* *Bacillus butanolivorans*	+ + + + + + + + +++ + + +	Biochemical testing and 16S rRNA gene sequencing	Cellulose powder Carboxymethylcellulose	Plowing and harrowing No-tillage All tillage treatments, including both with and without cover crop conditions Cereal cropping sequences consisting of five-member crop rotations: winter wheat (*Triticum aestivum* L.)- winter rape (*Brassica napus*)*-* spring wheat (*Triticum aestivum* L.)- spring barley (*Hordeum vulgare*)*-* pea (*Pisum sativum*)	Qualitative: Testing with Congo red and Gram’s iodine solution Quantitative: Activities of endoglucanases and β-glucosidase were determined in 200 mM MES (morpholineptansulfonic acid) solution. The hydrolytic activities were quantified using 4-methylumbelliferyl and 7-amino-4-methylcoumarin as fluorogenic conjugated substrates	[33]
Sugar cane (*Saccharum officinarum* L.)	Santa Rita, Brazil	*Bacillus* sp. C1AC55.07	+++	16S rRNA gene sequencing	Carboxymethylcellulose	Not specified	Qualitative: Congo red assay Quantitative: CMCase was determined using the DNS method, which involved the quantification of glucose levels via spectrophotometry at a wavelength of 540 nm	[121]
Sugar cane (*Saccharum officinarum* L.)	Manabi, Ecuador	*Bacillus* sp. AO-19	+++	Morphological testing	Carboxymethylcellulose	A decade-old sugarcane monoculture with substantial fibrous waste accumulation	Qualitative: Congo red assay	[122]
Corn (*Zea mays* L.)	Hungary	*Flavobacterium hungaricum* sp. *nov*	Not quantified	16S rRNA gene sequencing	Carboxymethylcellulose	The authors only mention that the soil pH was moderately alkaline and the soil was fertilized	Qualitative: Congo red assay	[123]
Alfalfa (*Medicago sativa* L.) Chard (*Beta vulgaris* L.) Grass (*Arrhenatherum elatius* (Linn.) Pressl) Common sorrel (*Rumex crispus* L.)	Mexico City, Mexico	*Microbacterium oxydans* *Streptomyces anulatus* *Cellulomonas cellulans* *Agrobacterium rubi* *Sphingobium-bacterium* *Alcaligenes* sp. *Pseudomonas pseudoalcaligenes* *Stenotrophomonas maltophilia* *Pseudomonas mendocina*	+++ +++ +++ ++ - - +++ +++ +++	16S rRNA gene sequencing *	Cellulose powder	The soils were collected from agricultural production systems known as *chinampas*. pH 8.0–8.6, 4.6% to 7.5% organic matter, loamy sandy clayey texture	Qualitative: Congo red assay	[35]
Rice (*Oryza sativa* L.)	Indonesia	*Bacillus stratosphericus* *Bacillus amyloliquefaciens* *Bacillus cereus* *Bacillus pumilus* *Citrobacter freundii* *Pseudomonas pseudoalcaligenes* *Rhodobacter aestuarii* *Bacillus marisflavi* *Pantoea dispersa* *Streptomyces coelicoflavus* *Pseudomonas mosselii* *Rhodococcus ruber* *Arthrobacter alpinus* *Streptomyces albidoflavus*	Not quantified	16S rRNA gene sequencing	Carboxymethylcellulose	Not specified	Qualitative: Congo red assay	[124]

* The sequences of the 16S rRNA gene have been archived in the GenBank database. + Level of cellulase activity as reported by the authors: +++ (highest activity), ++ (medium activity), + (low activity).

## Data Availability

Not applicable.

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
