# Peer review of "Cellulolytic Aerobic Bacteria Isolated from Agricultural and Forest Soils: An Overview"

_biology, 2024, doi:10.3390/biology13020102_

Round 1

Reviewer 1 Report

Comments and Suggestions for Authors

In my opinion, this work is a comprehensive qualitative review. The manuscript entitled "Cellulolytic aerobic bacteria isolated from agricultural and forest soils: A review" is of undoubted scientific value in the field of microbiology, including agricultural microbiology. The work opens prospects for the use of microorganisms with cellulolytic properties as bioorganic fertilisers, environmentally friendly and safe plant growth stimulants, which will contribute to the production of environmentally friendly agricultural products in general.

I have no significant comments. In my opinion, the manuscript is well structured. A sufficient amount of literature has been used.

A small clarification is needed in the sentence (line 82), as the scientific research is presented for more than a decade.

Author Response

Dear editor:

Regarding our manuscript with reference number biology-2839991, entitled “Cellulolytic Aerobic Bacteria Isolated from Agricultural and Forest Soils: An Overview,” we have adhered to all the valuable recommendations provided by the reviewers. The changes made to the manuscript have been highlighted in red. A detailed description of these changes and responses to the reviewers' inquiries are outlined below.

Reviewer 1.

In my opinion, this work is a comprehensive qualitative review. The manuscript entitled "Cellulolytic aerobic bacteria isolated from agricultural and forest soils: A review" is of undoubted scientific value in the field of microbiology, including agricultural microbiology. The work opens prospects for the use of microorganisms with cellulolytic properties as bioorganic fertilizers, environmentally friendly and safe plant growth stimulants, which will contribute to the production of environmentally friendly agricultural products in general.

I have no significant comments. In my opinion, the manuscript is well structured. A sufficient amount of literature has been used.

A small clarification is needed in the sentence (line 82), as the scientific research is presented for more than a decade.

Response: We greatly appreciate the reviewer's insights. In this updated manuscript, we have clarified that our literature review encompasses the past 11 years (L. 21, 23, 105). While review articles traditionally focus on literature from the most recent five years, we found that relevant publications from this extended time frame are relatively scarce. Nonetheless, this broader scope is justified by significant advancements in omics technologies over the past decade. The articles frequently cited during this period have been instrumental in shaping the content of our manuscript.

Reviewer 2 Report

Comments and Suggestions for Authors

Dear authors,

This review paper summarizes the topic of cellulolytic soil bacteria and relevant studies. The idea of the manuscript is interesting and has the potential to move the related research forward. However, I have several concerns about this manuscript requiring substantial revision. Please consider the comments and suggestions stated in the points below.

Simple summary

I find the part starting from the sentence "Cellulolytic bacteria have been isolated…" till the end of the simple summary (lines 22-25) unnecessary. On the other hand, the description of the review objectives is incomplete. Besides the information on cellulolytic bacteria from forest and agricultural soils, you also focused on some general aspects of cellulolytic soil bacteria and cellulases in your review paper.    

 Abstract

Firstly, in the abstract and mainly in the introduction, I miss the reason and motivation why you chose to write about the cellulolytic bacteria of forests and agricultural soils. It should be at least appointed why the agricultural and forest soils in the context of cellulolytic bacteria are important more or similarly compared to other biomes/environments (like grassland, desert, polar, urban or other soils). It is nicely justified in the first paragraphs of chapters 4 and 5, but a strong relationship between cellulose, forest/agricultural soils, and cellulolytic bacteria should be obvious first from the abstract and introduction.  

Lines 28-30: What was the key to creating an order (highest to lowest) of the most significant bacterial genera? Authors should change the word "representative" to "most abundant" or according to the idea of the ordering. Otherwise, it could be unclear to the reader. Please correct this in the abstract and chapters 4, 5 and 6. 

Lines 30 and 480: "Cellulolytic bacteria were selected by determining their hydrolysis capacity..." - Please reformulate to "Cellulolytic bacteria are selected..." or similar – it is a general approach, how to select them.

At the end of the abstract, the importance of the review should be stated. I mean, it could be advantageous to highlight the importance of cellulase(s), as you discuss in the second paragraph of the introduction, and the efforts to look for new cellulolytic strains among the soil microorganisms. This may catch the attention of your manuscript.

Note: Some comments below may also be reflected in the abstract. 

1. Introduction

The introduction should be better organized; it lacks flow and is a little hard to read. I suggest dividing the chapter into strictly focused parts or paragraphs dealing with a) the occurrence and distribution of lignocellulose and its components in the ecosystems and organisms, b) the characterization of cellulase and enzymatic activities (ECs), c) importance of cellulases, applications and microbial sources of the cellulolytic enzymes d) general notes about soil cellulolytic microorganisms (predominantly focused on bacteria), e) cultivability of soil bacteria, techniques applicable for their determination, f) importance of cellulolytic cellulose decomposition and cellulolytic bacteria in agricultural and forest soils, e) aims of the review

2. Importance and selection of cultivable cellulolytic bacteria in the soil

This chapter merges too many aspects of the topic, like the importance of cellulolytic soil bacteria and their role in cellulose degradation, crucial criteria for their selection (origin and identification) and determination of their activities (methods). Similarly, like in the introduction, I miss the flow of the text. Moreover, the previously mentioned aspects deserve more attention from the authors, in my opinion. Therefore, I suggest splitting this chapter into two comprehensive sections. One of the sections should be related to the detailed characteristics of cellulolytic soil bacteria and methods applicable to their identification. The second section could focus on the methods of the qualitative and quantitative analysis of cellulase. 

In particular, the identification methods are described very briefly, or neither. Of course, the methods for identifying cellulolytic strains could be genomic sequencing or proteomic tools, or 16S rRNA can be used to characterise microbial communities. However, genomic sequencing has the disadvantage that it may involve the cellulase-coding genes, which can be silent (cryptic), thus giving false positive results. From this point of view, the methods based on proteome analysis or transcriptomic or metatranscriptomic approaches working with the functional products of cellulase-coding genes are more promising. The metatranscriptomic approach was already used to characterise microbial communities of soil microorganisms and their cellulases in the millipede gut (see https://doi.org/10.3389/fevo.2022.931986). However, I think this approach can be used to characterise the communities or consortiums of cellulolytic cellulases and source cellulolytic bacteria present or isolated from agricultural or forest soils. Concerning the methods of the cellulase activity assessment, not only Congo Red or iodine staining (qualitative) or Bradford, Lowry, and DNS methods (quantitative) are applicable, but also other methods are usable for the analyses of cellulases, e.g. using FPU (filter- paper units) method (https://doi.org/10.1016/j.bej.2023.109196), 4-MUC assay (e.g, https://doi.org/10.1007/s11274-023-03620-5) or MS techniques (e.g., https://doi.org/10.1134/S0003683821100100).

Lines 118-119: "It is well-documented that most bacterial cellulases have better production at pH values of 6 to 9." - The statement " most bacterial cellulases have better production" is nonsense. Did you mean the bacterial production of cellulases or the activity of cellulases? You mixed two different parameters related to the cited research article. The first cited (ref 32) deals with the pH (6-9) for activity, while the second one (ref 33) deals with the pH (4-8) of cultivation conditions for cellulase production. Therefore, you probably meant for optimum enzymatic activity. However, you should also report the more recent studies on the effect of pH on cellulase production for different cellulolytic soil microorganisms. For instance, according to one research study (https://doi.org/10.1007/s11274-023-03620-5), not all bacteria are effective in producing cellulases at neutral pH. In this research, streptomycetes (+ fungi) showed significantly higher cellulase production at pH 5 than 7. This information may be especially interesting in the context of cellulolytic streptomycetes occurring in the soil of (coniferous) forests, which often have relatively low pH values.

3. Truly cellulolytic bacteria

Lines 170-172, line 35, lines 485-486: Please update the list of GH families connected with cellulolytic enzymes according to the most recent present in the CAZy database and cite the database directly. For instance, EC 3.2.1.14 activity is involved in GH124 and GH148 families, which are not stated in your manuscript.

Lines 222-224: Authors should explain and discuss in detail why cellulose is non-degradable despite its presence in the dead plant biomass. Of course, it is mainly due to the lack of GH48 family enzymes of most microbial decomposers. Another reason can represent the inaccessibility of cellulose in highly recalcitrant lignocellulosic composite polymer, which is encapsulated in and protected by a lignin skeleton. The breakdown of lignin is mediated by lignin-modifying enzymes, which could be scarce or less effective in soil bacteria than fungal LMEs. Moreover, the lignocellulosic structure should be properly described or visualized by the sole figure.

Chapters 4 and 5

In general, I find these two chapters, representing the most important parts of the manuscript, well processed. However, Tables 1 and 2 are too general and should be extended by providing the methods used to identify cellulolytic bacteria (16 rRNA seq, genomic seq, proteomic analysis) and cellulases analyses (qualitative and/or quantitative) used within each stated study. Also, related to the studies where the cellulolytic activities were assessed quantitively, you should at least provide the ranges of values of these activities and corresponding units. Or better, you could label and compare the intensity (e.g., in three grades, i.e. +, ++ or +++) of these activities for the listed genera/species within each relevant study. Besides that, it would also be beneficial to state (in the text or figures) the soil characteristics and composition (if available) within all the reported studies – important, especially in the context of what you mention in lines 326-329. This simple but detailed information provided in the tables should increase the attractiveness of the manuscript. 

In the points below, I also comment on some minor errors.

Lines 243-244: "These authors mentioned that they added the 243 genus Erwinia, which had not been previously reported as cellulolytic." – Please reformulate this sentence to "These authors reported, as the first, the evidence of the cellulolytic activity of the genus Erwinia." or similarly.

Line 254: Technically, bacterial isolate is not what you find in the environment. A bacterial isolate is a culture of a single species of bacteria obtained later in the lab from the collected (soil) samples. In the environment, you look for the new strains, species, genera (or some other taxon), so please substitute to "were obtained from these soils".

Line 386: Used the bacterial strain? For what? Be more specific, please.

Line 420: Describe "GH genes" rather than "genes encoding proteins related to GHs".

In the end, I like the last paragraph of Chapter 5 (lines 457-465). It nicely appoints the significance of soil cellulolytic microorganisms and shows why increasing the awareness of these microorganisms is so important. I would truly appreciate providing more of these ideas throughout your manuscript.

6. Conclusions and future directions

This chapter is too long, especially the "conclusions" part (lines 471-491). Moreover, it mostly repeats the statements from the previous chapters. It would be beneficial to re-write this
chapter - condense it, highlight the most important facts from the previous chapters and possibly add more of your ideas and opinions. Please consider also the certain points below. 

Lines 483-485: "However, the hydrolysis capacity does not always reflect the bacterial production of cellulolytic enzymes." – Hydrolysis capacity of cellulases or bacteria? Please specify it.

Lines 490-491: "In many of the examined studies, artificial cellulose substrates were utilized to evaluate bacterial hydrolysis capacity. Nevertheless, this method does not guarantee the precise identification of truly cellulolytic bacteria." – Please, since you write "this method" in the second sentence, you should specify which method was applied in the examined studies.

Line 493: As I stated previously, transcriptomic tools could also be crucial, maybe even more than genomic tools from a certain point of view, since they work with the RNA products of truly expressed genes translated to proteins (cellulases for our case).

Lines 500-502: "Finally, the potential of truly cellulolytic bacteria should be explored in different areas such as plant growth promotion as it could contribute significantly to plant growth and development" – Please reformulate or simplify this sentence – you state “plant growth” twice in one sentence.

Lines 502-504: "Alternatively, examining the variation, abundance, and diversity of truly cellulolytic microbial communities in the soil could serve as indicators of the quality of this crucial natural resource." – This sentence is unclear to me. What "crucial natural resource" do you mean? Cellulase, cellulose, microbial communities or something different? Furthermore, you should describe what quality parameters are affected by the named characteristics of microbial communities.

Comments on the Quality of English Language

The quality of the English language could be better. I recommend professional language editing to avoid some issues related to the flow of the text, which is disordered and inconsequential in some chapters (caused by mixing more topics within single paragraphs or missing segues) and stylistic errors (unclear meaning of some sentences, incorrect words). The language editing would certainly improve the readability of the manuscript.

Author Response

Dear editor:

Regarding our manuscript with reference number biology-2839991, entitled “Cellulolytic Aerobic Bacteria Isolated from Agricultural and Forest Soils: An Overview,” we have adhered to all the valuable recommendations provided by the reviewers. The changes made to the manuscript have been highlighted in red. A detailed description of these changes and responses to the reviewers' inquiries are outlined below.

Reviewer 2.

This review paper summarizes the topic of cellulolytic soil bacteria and relevant studies. The idea of the manuscript is interesting and has the potential to move the related research forward. However, I have several concerns about this manuscript requiring substantial revision. Please consider the comments and suggestions stated in the points below.

Simple summary

I find the part starting from the sentence "Cellulolytic bacteria have been isolated…" till the end of the simple summary (lines 22-25) unnecessary.

Response: We are grateful for the reviewer's insightful comment. In response, the specified sentence has been removed in this revised version of the manuscript.

On the other hand, the description of the review objectives is incomplete. Besides the information on cellulolytic bacteria from forest and agricultural soils, you also focused on some general aspects of cellulolytic soil bacteria and cellulases in your review paper.

Response: In recognition of the reviewer's pertinent observation, we have meticulously revised the manuscript to delineate the study’s objectives more precisely. This revision incorporates a comprehensive discussion on the fundamental characteristics of cellulolytic soil bacteria, with a particular focus on the enzymes they produce, namely cellulases. This is detailed in lines 20-21, 23-24, and 105-106 of the manuscript.

Abstract

Firstly, in the abstract and mainly in the introduction, I miss the reason and motivation why you chose to write about the cellulolytic bacteria of forests and agricultural soils. It should be at least appointed why the agricultural and forest soils in the context of cellulolytic bacteria are important more or similarly compared to other biomes/environments (like grassland, desert, polar, urban or other soils). It is nicely justified in the first paragraphs of chapters 4and 5, but a strong relationship between cellulose, forest/agricultural soils, and cellulolytic bacteria should be obvious first from the abstract and introduction.

Response: We are grateful for the valuable feedback provided by the reviewer. Consequently, we have incorporated a rationale explaining our decision to undertake a review on cellulolytic bacteria present in forest and agricultural soils (L. 25-27 in the abstract; L. 48-52 in the introduction).

Lines 28-30: What was the key to creating an order (highest to lowest) of the most significant bacterial genera? Authors should change the word "representative" to "most abundant" or according to the idea of the ordering. Otherwise, it could be unclear to the reader. Please correct this in the abstract and chapters 4, 5 and 6.

Response: The bacterial genera have been organized in descending order based on their reported frequency. Following the reviewer's suggestion, we have substituted 'representative' with 'most abundant' to reflect their prevalence more accurately.

Lines 30 and 480: "Cellulolytic bacteria were selected by determining their hydrolysis capacity..." - Please reformulate to"Cellulolytic bacteria are selected..." or similar – it is a general approach, how to select them.

Response: This suggestion has been incorporated in this corrected version.

At the end of the abstract, the importance of the review should be stated. I mean, it could be advantageous to highlight the importance of cellulase(s), as you discuss in the second paragraph of the introduction, and the efforts to look for new cellulolytic strains among the soil microorganisms. This may catch the attention of your manuscript.

Note: Some comments below may also be reflected in the abstract.

Response: As suggested by the reviewer, the importance of cellulases and efforts to search for new cellulolytic strains among soil microorganisms have been included at the end of the abstract (L. 37-40).

  1. Introduction



The introduction should be better organized; it lacks flow and is a little hard to read. I suggest dividing the chapter into strictly focused parts or paragraphs dealing with a) the occurrence and distribution of lignocellulose and its components in the ecosystems and organisms, b) the characterization of cellulase and enzymatic activities (ECs), c) importance of cellulases, applications and microbial sources of the cellulolytic enzymes d) general notes about soil cellulolytic microorganisms (predominantly focused on bacteria), e) cultivability of soil bacteria, techniques applicable for their determination, f) importance of cellulolytic cellulose decomposition and cellulolytic bacteria in agricultural and forest soils, e) aims of the review

Response: The introduction was reorganized based on the reviewer's comments.

  1. Importance and selection of cultivable cellulolytic bacteria in the soil

This chapter merges too many aspects of the topic, like the importance of cellulolytic soil bacteria and their role in cellulose degradation, crucial criteria for their selection (origin and identification) and determination of their activities (methods). Similarly, like in the introduction, I miss the flow of the text. Moreover, the previously mentioned aspects deserve more attention from the authors, in my opinion. Therefore, I suggest splitting this chapter into two comprehensive sections. One of the sections should be related to the detailed characteristics of cellulolytic soil bacteria and methods applicable to their identification. The second section could focus on the methods of the qualitative and quantitative analysis of cellulase.

In particular, the identification methods are described very briefly, or neither. Of course, the methods for identifying cellulolytic strains could be genomic sequencing or proteomic tools, or 16S rRNA can be used to characterise microbial communities. However, genomic sequencing has the disadvantage that it may involve the cellulase-coding genes, which can be silent (cryptic), thus giving false positive results. From this point of view, the methods based on proteome analysis or transcriptomic or metatranscriptomic approaches working with the functional products of cellulase-coding genes are more promising. The metatranscriptomic approach was already used to characterise microbial communities of soil microorganisms and their cellulases in the millipede gut (seehttps://doi.org/10.3389/fevo.2022.931986). However, I think this approach can be used to characterise the communities or consortiums of cellulolytic cellulases and source cellulolytic bacteria present or isolated from agricultural or forest soils. Concerning the methods of the cellulase activity assessment, not only Congo Red or iodine staining (qualitative) or Bradford, Lowry, and DNS methods (quantitative) are applicable, but also other methods are usable for the analyses of cellulases, e.g. using FPU (filter- paper units) method(https://doi.org/10.1016/j.bej.2023.109196), 4-MUC assay (e.g, https://doi.org/10.1007/s11274-023-03620-5) or MS techniques(e.g., https://doi.org/10.1134/S0003683821100100).

Response: The chapter has been divided into two sections as suggested by the reviewer. One section is focused on "Soil cellulolytic bacteria and methods for their identification" (L. 109-151). The other section addresses" Methods for qualitative and quantitative determination of cellulase activity" (L. 152-246).

Lines 118-119: "It is well-documented that most bacterial cellulaseshave better production at pH values of 6 to 9." - The statement "most bacterial cellulases have better production" is nonsense. Didyou mean the bacterial production of cellulases or the activity of cellulases? You mixed two different parameters related to the citedresearch article. The first cited (ref 32) deals with the pH (6-9) foractivity, while the second one (ref 33) deals with the pH (4-8) ofcultivation conditions for cellulase production. Therefore, youprobably meant for optimum enzymatic activity. However, youshould also report the more recent studies on the effect of pH oncellulase production for different cellulolytic soil microorganisms.For instance, according to one research study(https://doi.org/10.1007/s11274-023-03620-5), not all bacteria areeffective in producing cellulases at neutral pH. In this research,streptomycetes (+ fungi) showed significantly higher cellulaseproduction at pH 5 than 7. This information may be especiallyinteresting in the context of cellulolytic streptomycetes occurring inthe soil of (coniferous) forests, which often have relatively low pHvalues.

Response: The reviewer's feedback is highly appreciated. Considering that cellulolytic microorganisms exhibit optimal expression in a culture medium when incubated under conditions like their natural environment (Goyari et al., 2014), we have opted to omit this paragraph to prevent any potential confusion. In this revised edition, we include findings from Koubová et al. (2023), which indicate that bacteria from the Streptomyces genus show markedly increased cellulase production at a pH of 5, in contrast to a pH of 7 (L. 243-246).

  1. Truly cellulolytic bacteria

Lines 170-172, line 35, lines 485-486: Please update the list of GH families connected with cellulolytic enzymes according to the most recent present in the CAZy database and cite the database directly. For instance, EC 3.2.1.14 activity is involved in GH124 and GH148families, which are not stated in your manuscript.

Response: The list of GH families related to cellulolytic enzymes has been updated, and the CAZy database was directly cited (L. 33-34; 284-285).

Lines 222-224: Authors should explain and discuss in detail why cellulose is non-degradable despite its presence in the dead plant biomass. Of course, it is mainly due to the lack of GH48 family enzymes of most microbial decomposers. Another reason can represent the inaccessibility of cellulose in highly recalcitrant lignocellulosic composite polymer, which is encapsulated in and protected by a lignin skeleton. The breakdown of lignin is mediated by lignin-modifying enzymes, which could be scarce or less effective in soil bacteria than fungal LMEs. Moreover, the lignocellulosic structure should be properly described or visualized by the sole figure.

Response: We have included a detailed explanation concerning the recalcitrance of cellulose to enzymatic hydrolysis (refer to lines 252-254 and 257-273). Additionally, Figure 1 has been enhanced to depict the crystalline structure of cellulose more accurately.

Chapters 4 and 5

In general, I find these two chapters, representing the mos timportant parts of the manuscript, well processed. However, Tables1 and 2 are too general and should be extended by providing the methods used to identify cellulolytic bacteria (16 rRNA seq, genomic seq, proteomic analysis) and cellulases analyses (qualitative and/or quantitative) used within each stated study. Also, related to the studies where the cellulolytic activities were assessed quantitively, you should at least provide the ranges of values of these activities and corresponding units. Or better, you could label and compare the intensity (e.g., in three grades, i.e. +, ++ or +++) of these activities for the listed genera/species within each relevant study. Besides that, it would also be beneficial to state (in the text or figures) the soil characteristics and composition (if available) within all the reported studies – important, especially in the context of what you mention in lines 326-329. This simple but detailed information provided in the tables should increase the attractiveness of the manuscript.

Response: We are grateful for the reviewer's valuable suggestion. In response, we have augmented the tables to include not only the methods employed for identifying cellulolytic bacteria but also the analyses of their cellulase activities. Furthermore, we have updated the manuscript text to encompass discussions of soil characteristics as described in the reviewed studies.

In the points below, I also comment on some minor errors.

Lines 243-244: "These authors mentioned that they added the genus Erwinia, which had not been previously reported as cellulolytic." – Please reformulate this sentence to "These authors reported, as the first, the evidence of the cellulolytic activity of the genus Erwinia." or similarly.

Response: This correction has been incorporated (L. 359-360).

Line 254: Technically, bacterial isolate is not what you find in the environment. A bacterial isolate is a culture of a single species of bacteria obtained later in the lab from the collected (soil) samples. In the environment, you look for the new strains, species, genera (or some other taxon), so please substitute to "were obtained from these soils".

Response: Thank you for the comment. This correction has been implemented (L. 370-371).

Line 386: Used the bacterial strain? For what? Be more specific, please.

Response: We have indicated that the Bacillus sp. strain C1AC55.07 was used for the optimization of cellulase production (L. 503-504).

Line 420: Describe "GH genes" rather than "genes encoding proteins related to GHs".

Response: This correction has been incorporated (L. 534).

In the end, I like the last paragraph of Chapter 5 (lines 457-465). It nicely appoints the significance of soil cellulolytic microorganisms and shows why increasing the awareness of these microorganisms is so important. I would truly appreciate providing more of these ideas throughout your manuscript.

Response: Additional insights regarding the importance of cellulolytic microorganisms have been included (L. 120-124).

  1. Conclusions and future directions

This chapter is too long, especially the "conclusions" part (lines471-491). Moreover, it mostly repeats the statements from the previous chapters. It would be beneficial to re-write this chapter - condense it, highlight the most important facts from the previous chapters and possibly add more of your ideas and opinions. Please consider also the certain points below.

Response: We sincerely appreciate the reviewer's valuable suggestions. In accordance with this guidance, we have thoroughly revised the conclusions to underscore the points we deem most critical, reflecting a comprehensive review of the topics addressed.

Lines 483-485: "However, the hydrolysis capacity does not always reflect the bacterial production of cellulolytic enzymes." –Hydrolysis capacity of cellulases or bacteria? Please specify it.

Response: This sentence was removed because the conclusions were rewritten.

Lines 490-491: "In many of the examined studies, artificial cellulose substrates were utilized to evaluate bacterial hydrolysis capacity. Nevertheless, this method does not guarantee the precise identification of truly cellulolytic bacteria." – Please, since you write "this method" in the second sentence, you should specify which method was applied in the examined studies.

Response: We have rewritten the conclusions and now specify that qualitative evaluation does not guarantee the accurate identification of truly cellulolytic bacteria (L. 584-585).

Line 493: As I stated previously, transcriptomic tools could also be crucial, maybe even more than genomic tools from a certain point of view, since they work with the RNA products of truly expressed genes translated to proteins (cellulases for our case).

Response: The reviewer's comment is very important. We have included that transcriptomic tools could be potentially useful in characterizing communities of cellulolytic bacteria present in, or isolated from, agricultural or forest soils (L. 586-588).

Lines 500-502: "Finally, the potential of truly cellulolytic bacteria should be explored in different areas such as plant growth promotion as it could contribute significantly to plant growth and development" – Please reformulate or simplify this sentence – you state “plant growth” twice in one sentence.

Response: The wording of the specified sentence has been carefully revised and enhanced for clarity and precision (L. 592-594).

Lines 502-504: "Alternatively, examining the variation, abundance, and diversity of truly cellulolytic microbial communities in the soil could serve as indicators of the quality of this crucial natural resource." – This sentence is unclear to me. What "crucial natural resource" do you mean? Cellulase, cellulose, microbial communities or something different? Furthermore, you should describe what quality parameters are affected by the named characteristics of microbial communities.

Response: The wording of the specified sentence has been carefully revised and enhanced for clarity and precision (L. 594-595).

The quality of the English language could be better. I recommend professional language editing to avoid some issues related to the flow of the text, which is disordered and inconsequential in some chapters (caused by mixing more topics within single paragraphs or missing segues) and stylistic errors (unclear meaning of some sentences, incorrect words). The language editing would certainly improve the readability of the manuscript.

Response: The quality of the English throughout the manuscript has been reviewed and improved.

Reviewer 3 Report

Comments and Suggestions for Authors

Dear authors, I have carefully read your review of the scientific studies published over the past ten years about cultivable cellulolytic bacteria reported in forest and agricultural soils of different regions of the world.

I have screened research articles and reviews in this scientific field using search of Scopus and Google scholar. As a result, I did not find a review identical to the article being evaluated.  Several reviews relate to cellulolytic bacteria of mangrove forests, genes of cellulolytic enzymes, composting of agricultural residues, etc. Most of these reviews are cited in the manuscript. The proposed topic, layout and purpose of the new review, as I understood them, seem original.

The cited articles have been published in the last 11 years. This exceeds the usually recommended period of five years. But it seems justified to me because the widespread use of omics methods covers exactly the last decade. Most of the frequently cited articles during this period were used to write this manuscript. So modern research directions and achievements in the study of the diversity of agricultural and forest cellulolytic bacteria are reflected correctly. The statements and conclusions made are supported by the listed citations.

The text of the article is clear and structured. The drawings are colorful. But it confuses me that the meaning of Figure 1 is difficult to understand without reading the text of the article.

As I understood it, the knowledge gap lies in the insufficient number of omic studies of forest and agricultural cellulolytic bacteria in order to assess their diversity and sort them according to the bioavailability of cellulose substrates. This opinion has been expressed by scientists not for the first time in the last decade. But omics research is not done as often in the ecology of soil microorganisms as in medicine, and gaps still remain. It would be interesting to analyze, based on the papers cited in sections 3 and 4, what progress has been made in this area in recent years.  It seems to me that it would be good to write in Tables 1 and 2, in addition to the substrates for isolation, which modern methods were used to study the listed strains of cellulolytic bacteria. It is also interesting whether the complete genomes or enzymes of recently isolated bacteria have been placed in open databases.

Author Response

Dear editor:

Regarding our manuscript with reference number biology-2839991, entitled “Cellulolytic Aerobic Bacteria Isolated from Agricultural and Forest Soils: An Overview,” we have adhered to all the valuable recommendations provided by the reviewers. The changes made to the manuscript have been highlighted in red. A detailed description of these changes and responses to the reviewers' inquiries are outlined below.

Reviewer 3.

Dear authors, I have carefully read your review of the scientific studies published over the past ten years about cultivable cellulolytic bacteria reported in forest and agricultural soils of different regions of the world. I have screened research articles and reviews in this scientific field using search of Scopus and Google scholar. As a result, I did not find a review identical to the article being evaluated.

Several reviews relate to cellulolytic bacteria of mangrove forests, genes of cellulolytic enzymes, composting of agricultural residues, etc. Most of these reviews are cited in the manuscript. The proposed topic, layout, and purpose of the new review, as I understood them, seem original.

The cited articles have been published in the last 11 years. This exceeds the usually recommended period of five years. But it seems justified to me because the widespread use of omics methods covers exactly the last decade. Most of the frequently cited articles during this period were used to write this manuscript. So modern research directions and achievements in the study of the diversity of agricultural and forest cellulolytic bacteria are reflected correctly.

The statements and conclusions made are supported by the listed citations.

The text of the article is clear and structured. The drawings are colorful. But it confuses me that the meaning of Figure 1 is difficult to understand without reading the text of the article.

As I understood it, the knowledge gap lies in the insufficient number of omic studies of forest and agricultural cellulolytic bacteria in order to assess their diversity and sort them according to the bioavailability of cellulose substrates. This opinion has been expressed by scientists not for the first time in the last decade. But omics research is not done as often in the ecology of soil microorganisms as in medicine, and gaps still remain. It would be interesting to analyze, based on the papers cited in sections 3 and4, what progress has been made in this area in recent years.

Response: We sincerely appreciate your valuable suggestions. Recent studies that employ genome or transcriptome sequencing, in conjunction with proteome or metabolome analysis, have marked a new era in understanding the role of plant cell wall-degrading enzymes in insect digestive physiology (Tokuda, 2019). This aspect is now briefly outlined in lines 135-151. Additionally, Figure 1 has been enhanced to facilitate better comprehension.

It seems to me that it would be good to write in Tables 1 and 2, in addition to the substrates for isolation, which modern methods were used to study the listed strains of cellulolytic bacteria.

It is also interesting whether the complete genomes or enzymes of recently isolated bacteria have been placed in open databases.

Response: We express our gratitude for the reviewer's valuable suggestion. Accordingly, we have now included in the tables both the methodologies employed for the identification of cellulolytic bacteria and the analyses of their cellulase activities.

Furthermore, within the tables, an asterisk (*) is used to denote instances where complete genomes have been deposited in the GenBank database.

Additional comments:

In this section, please provide details regarding where data supporting reported results can be found, including links to publicly archived datasets analyzed or generated during the study. Please refer to suggested Data Availability Statements in section “MDPI Research Data Policies” at https://www.mdpi.com/ethics. You might choose to exclude this statement if the study did not report any data.

Response: We believe that all data utilized for the manuscript's preparation are included in the works cited in the references section.

Round 2

Reviewer 2 Report

Comments and Suggestions for Authors

After correcting all critical issues,  I have no further comments or suggestions, and I recommend your manuscript for publication in its present form.

Reviewer 3 Report

Comments and Suggestions for Authors

Dear authors, I have read the revised version of the article. All the changes made to the manuscript seemed appropriate to me.